# Effects on health-related quality of life in the randomized, controlled crossover trial ADIRA (Anti-inflammatory Diet In Rheumatoid Arthritis)

**Anna Turesson Wadell**[1]*, **Linnea Bärebring**[1], **Erik Hulander**[1], **Inger Gjertsson**[2], **Lars Hagberg**[3], **Helen M. Lindqvist**[1], **Anna Winkvist**[1]*

1 Department of Internal Medicine and Clinical Nutrition, Institute of Medicine, Sahlgrenska Academy, University of Gothenburg, Gothenburg, Sweden, 2 Department of Rheumatology and Inflammation Research, Institute of Medicine, Sahlgrenska Academy, University of Gothenburg, Gothenburg, Sweden, 3 Centre for Health Care Science, Faculty of Medicine and Health, Örebro University, Örebro, Sweden

* anna.turesson.wadell@gu.se (ATW); anna.winkvist@nutrition.gu.se (AW)

**Data Availability Statement:** Data cannot be made freely available as they are subject to secrecy in accordance with the Swedish Public Access to

## Abstract

### Background

Patients with Rheumatoid Arthritis (RA) often report impaired health-related quality of life (HrQoL) such as difficulties in daily life, pain, fatigue and an affected social life. Even when lowering disease activity, pharmacological treatment does not always resolve these factors.

### Objective

To investigate if a proposed anti-inflammatory diet improves HrQoL in patients with RA.

### Design

In this controlled crossover trial, 50 patients were randomized to start with either an intervention diet (anti-inflammatory) or a control diet (usual Swedish intake) for ten weeks followed by a wash out period before switching to the other diet. Participants received food equivalent to ~1100 kcal/day, five days/week, and instructions to consume similarly for the remaining meals. HrQoL was evaluated using Health Assessment Questionnaire (HAQ), 36-item Short Form Survey (SF-36), Visual Analogue Scales (VAS) for pain, fatigue and morning stiffness, and a time scale for morning stiffness.

### Results

Forty-seven participants completed ≥1 diet period and were included in the main analyses. No significant difference between intervention and control diet at end of diet periods was observed for any outcome. However, significant improvements were obtained for SF-36 Physical Functioning (mean:5.79, SE: 2.12, 95% CI: 1.58, 10.01) during the intervention diet period. When excluding participants with anti-rheumatic medication changes, the differences between diet periods increased for most outcomes, favoring the intervention diet

Information and Secrecy Act [Offentlighets- och sekretesslagen, OSL, 2009:400], but can be made available to researchers upon request (subject to a review of secrecy). Requests for data should be made to the head of the Department of Internal Medicine and Clinical Nutrition at the University of Gothenburg, Sweden, Jorgen.isgaard@medic.gu.se. Requests could also be addressed Anna Winkvist, anna.winkvist@nutrition.gu.se.

**Funding:** Granted by funds from the Swedish government under the ALF agreement (ALFGBG-74630) (AW), Swedish Research Council for Health, Working Life and Welfare (FORTE https://forte.se/en/) (AW), the Lennander Foundation, Sahlgrenska University Hospital Foundations https://www.uaf.se/en/foundation/?s=91512#__utma=1.2945126.1597479581.1601631581.1601631613.3&__utmb=1.1.10.1601631613&__utmc=1&__utmx=-&__utmz=1.1601631613.3.2.utmcsr=google|utmccn=(organic)|utmcmd=organic|utmctr=(not%20provided)&__utmv=-&__utmk=152169273 (LB), the Inger Bendix Foundation https://www.stiftelsemedel.se/inger-bendix-stiftelse-fr-medicinsk-forskning/ (AW) and the Gothenburg Region Foundation for Rheumatology Research (GSFR) (LB). The funders had no role in study design, data collection and analysis, decision to publish, or preparation of the manuscript.

**Competing interests:** The authors have declared that no competing interests exist.

period, and the difference for SF-36 Physical Functioning became significant (n = 25, mean:7.90, 95% CI:0.56, 15.24, p = 0.036).

## Conclusions

In main analyses, the proposed anti-inflammatory diet did not significantly improve HrQoL for patients with RA compared to control diet. In sub-analyses, significant improvements in physical functioning were detected. Larger studies with consistent medication use and in populations more affected by the disease may be needed to obtain conclusive evidence.

## Introduction

Rheumatoid Arthritis (RA) is a chronic autoimmune disease primarily affecting the joints, and that also is associated with increased risk for comorbidities such as cardiovascular diseases [1]. The inflammation in the synovial membrane of the joint and the subsequent destruction of cartilage and bone cause pain and stiffness, and eventually lead to reduced functional ability. As a result, difficulties in daily living such as dressing, grooming, walking and cooking are common. In addition, pain and fatigue may limit the ability to work and often has a negative impact on social life. Consequently, a lower health-related quality life (HrQoL) is often reported in patients with RA [2].

Several nutrients, foods and diets have been of interest among RA researchers, clinicians and patients for a long time [3]. In previous trials, the Mediterranean diet resulted in higher functional ability and less pain [4,5], reduced morning stiffness [5] and an improvement in vitality [4]. A meta-analysis including randomized controlled trials investigating supplemental n-3 fatty acids in RA also confirmed some of these beneficial effects [6]. Similarly, a higher seafood intake has been associated with a higher functional ability [7] and mental health [8]. Probiotic as well as synbiotic supplementation resulted in less pain compared to placebo [9,10] and in one study, within-group differences showed an improved physical ability [11]. Prebiotics in the form of high-fiber bars also showed potential to aspects of HrQoL [12]. Supplements with fruit extract, quercetin, sesamin and cinnamon, rich in vitamins with antioxidant effects and/or phenolic compounds, improved physical ability and lowered both pain and morning stiffness duration [13–19]. In addition, a longer duration of morning stiffness has been associated with a lower overall dietary quality [20]. However, although significant results, the quality of some of these studies can be questioned, and to date, there are no disease-specific dietary recommendations; patients with RA are referred to the same dietary advice as the general population. Nevertheless, the potential of nutrients and foods to reduce symptoms and act as adjuvant therapy remains and warrants investigation in larger well-conducted trials.

In 2017–2018 the crossover trial ADIRA (Anti-inflammatory Diet In Rheumatoid Arthritis) was performed. The primary aim of the trial was to evaluate effects on disease activity from a diet consisting of foods with proposed anti-inflammatory properties, i.e. rich in dietary fiber (prebiotics), n-3 fatty acids, probiotics and antioxidants, compared to a diet nutritionally similar to the average Swedish dietary intake. Although no significant differences between the diets were demonstrated in the main analysis, Disease Activity Score-28 (DAS28) was significantly lower after intervention period compared to before intervention period, and also compared to control period in unadjusted analyses [21].

The primary goal for current pharmacological treatment, e.g. the immunosuppressive Disease Modifying Anti-Rheumatic Drugs (DMARD's), is to lower the disease activity in order to

reduce inflammation and improve both joint status and self-reported wellbeing, and achieve remission [1]. However, even when the criteria for remission are fulfilled, many patients still experience debilitating symptoms like pain and fatigue [22,23]. Thus, there is a pressing need to identify other treatment options to complement pharmacological treatment to improve disease related HrQoL.

Patient-reported outcomes (PROs) are reports "[. . .] of the status of a patient's health condition that comes directly from the patient, without interpretation of the patient's response by a clinician or anyone else" [24, p.2]. In the present study, we aimed to investigate possible effects of the ADIRA diet on functional ability and other aspects of HrQoL by using PRO instruments.

## Subjects and methods

### The ADIRA trial

ADIRA was a single-blinded, randomized, controlled crossover trial. Details about study design have been described in previous publications [21,25]. Briefly, patients with RA, 18–75 years old, with a disease duration of ≥2 years and living in the Gothenburg area, Sweden, were identified from the Swedish Rheumatology Quality register (SRQ). Thereafter, the research groups sent the patients a letter with an enquiry to participate in the study and those interested were invited to a screening visit. Participants were included if the disease was active at screening (DAS28-ESR ≥2.6) and their anti-rheumatic medication had not been changed the preceding 8 wks. Pregnancy or life-threatening diseases, allergies or intolerances to non-exchangeable study food as well as inability to understand the study information, were all exclusion criteria.

The trial was performed in two batches and all study visits took place at the Clinical Rheumatology Research Centre, Clinic of Rheumatology, Sahlgrenska University Hospital, Gothenburg, Sweden.

The trial was registered at ClinicalTrials.gov, no NCT02941055 https://clinicaltrials.gov/show/NCT02941055.

### Dietary intervention

A computer-generated randomization list allocated participants 1:1 to start with either the intervention or control diet (see details about the diets below). Participants maintained the diet for 10 wks followed by a washout period for a median of 4 mo. Thereafter diet regimen was switched. Food corresponding to ~1100 kcal/d (one breakfast, one main meal and one snack, 5 d/wk, during both diet periods) were delivered weekly to participants' homes using the home delivery food chain *mat.se*. This food was added to their usual diet, which was modified to suit the respective diets described below.

Participants were instructed to abstain from all nutritional supplements during the study, except for those prescribed by a physician. They were also instructed to note all changes in pharmacological treatment. In an effort to blind participants, the intervention diet was called "fiber diet" and the control diet "protein diet" and it was communicated to the participants that effects of both these diets were investigated. Additionally, the nurses who conducted the examinations were blinded to participant allocation. In each diet period, the participants were contacted by phone once mid-period for compliance control. Compliance was defined as having consumed ≥80% of the food provided the week before the phone call [21].

**Intervention diet.** The anti-inflammatory diet was a portfolio diet including foods with suggested anti-inflammatory properties and food components shown to have promising effects on RA symptoms as reported by previous research. A thorough description of the diet has

been presented previously [21,25]. Briefly, breakfast contained whole grain, low fat dairy, fruits or berries, nuts (mainly walnuts) and a juice shot containing probiotics (*Lactobacillus plantarum v299*). The main meals were composed of fatty fish or legumes, potatoes or grains (predominantly whole grain), vegetables and low fat dairy. As snacks, participants received two fruits per day. In total, i.e. including meals not provided by the study, participants were instructed to limit meat consumption to ≤3 times/wk, consume ≥5 portions/d of fruit, berries and vegetables, to choose low fat dairy and whole grain, as well as to use oil or margarine as cooking fat.

**Control diet.**   It was intended for the total dietary intake during the control period to nutritionally correspond to an average Swedish dietary intake [26]. Participants received breakfast composed of orange juice and a mix of yoghurt and quark served with cornflakes or white bread with butter and cheese. The main meals were composed of meat or chicken, potatoes or white rice and high fat dairy. Quark, protein pudding and protein bars were given as snacks. In total, participants were instructed to consume meat ≥5 times/wk, limit seafood intake to ≤1 time/wk and fruit and vegetables to ≤5 portions/day, to choose high fat dairy, use butter as cooking fat and to abstain from probiotics.

## Dietary assessment and lifestyle questionnaire

The participants performed a Food Frequency Questionnaire (FFQ) and a lifestyle questionnaire at the screening visit, which are both fully described elsewhere [21].

The FFQ was used to evaluate the participants' dietary quality before study start. An index created by the Swedish National Food Agency [27] was calculated from the FFQ, and the scores categorized as *poor* (0–4 points), *fair* (5–8 points) or *high dietary quality* (9–12 points) [28].

The lifestyle questionnaire contained questions about lifestyle and sociodemographic factors, e.g. educational level (primary school, 2-y upper secondary school, 3-y upper secondary school, university degree or equal, or no education), occupational status (does not work, <15 h/wk, 16–30 h/wk, 31–40 h/wk, or >40 h/wk, parent's birth place (Europe, Middle East, Africa, Asia, North America, or South America) and nicotine use (yes [cigarettes, chewing gum, snuff or nicotine patches] or no).

## Outcome variables

To assess functional ability and other aspects of HrQoL, self-administered questionnaires and Visual Analogue Scales (VAS) were used. The assessments were performed at all four study visits (i.e. before and after each diet period). All questionnaires were harmonized in layout, leading to minor changes from their original versions.

**Health assessment questionnaire.**   To evaluate functional ability, a Swedish version of the self-administered Stanford Health Assessment Questionnaire (HAQ), validated for patients with RA, was used [29,30]. The questionnaire contains 20 items regarding daily living over the past week, involving eight domains: *dressing and grooming*, *arising*, *eating*, *walking*, *hygiene*, *reach*, *grip* and *activities*. The six possible answers range from *Without any difficulty* through to *Unable to do*, with two of them addressing the need for aiding devices or assistance from another person. The final score (HAQ Disability Index [HAQ-DI]), which is an average of the eight domains, ranges from 0 to 3 where 3 indicates complete disability.

Three participants had left one item each unanswered, seven participants had double responses to one item each, and for one participant, the whole form was missing. For single items left unanswered at the third visit, the same response as for the first visit was imputed. There were no missing data at the other visits. For participants with double responses to an

item, with one of them being *need for aiding*, the latter was chosen. If a whole form was missing at the third visit, the same score as for the first visit was imputed. No whole forms were missing at the other visits.

**SF-36v2Ⓡ.**   The SF-36v2Ⓡ standard (4-week recall) form is a widely used questionnaire containing 36 items about functional ability and well-being in physical, social and mental aspects reflecting the past four weeks [31,32]. SF-36 has been shown to be a valid measure of HrQoL in patients with RA [33] and the Swedish version has been validated in a random sample of the Swedish adult population [34,35]. Thirty-five of the questions are divided into eight health domains: *Physical functioning (PF)*, *Role limitations due to physical problems (Role-Physical, RP)*, *Bodily pain (BP)*, *General health (GH)*, *Vitality (VT)*, *Social Functioning (SF)*, *Role limitations due to emotional problems (Role-Emotional, RE)* and *Mental health (MH)*. Each of these scales ranges from 0–100 where 100 is considered the best health state. The scales can then be summarized into two summary scores: Physical Component Summary (PCS) and Mental Component Summary (MCS).

To calculate the score for each domain and summary scores, OptumⓇ PRO CoRE Smart MeasurementⓇ System Version 1.5.7240.26936 (Optum, Inc) with the Missing Data Estimation Method, was used. Three participants had left one item each unanswered, and two participants had each marked two responses not adjacent to each other to one item. These items were considered missing and handled through the software program.

**Pain, fatigue and morning stiffness.**   To evaluate the participants' pain, fatigue and morning stiffness, a VAS with the following question was used: *During last week, how much pain/fatigue/morning stiffness at wake-up have you suffered from because of your rheumatic disease*? The participants were instructed to mark their response on a 100 mm line ranging from *No pain/fatigue/stiffness* to *Worst imaginable pain/fatigue/stiffness*. For one participant, the response to the VAS morning stiffness was missing at the fourth visit, and so the same value as for the second visit was imputed.

Participants also indicated duration of morning stiffness on a scale created for this trial, which was graded in intervals of fifteen minutes, from 0 hours through >2 ½ hours. If the mark was in between two interval ticks, 5, 7.5 or 10 minutes were added to the previous interval. Results are presented in minutes with a maximum of 150 minutes (corresponding to >2 ½ hours).

### Other clinical assessments

At screening, height was measured with a wall-mounted stadiometer to the closest 0.5 cm. Weight was measured at all visits. Participants were weighed in light clothing and without shoes and therefore 1 kg was subtracted from the weight. If unable to remove shoes, 1.5 kg was subtracted. Body Mass Index (BMI) was calculated as $kg/m^2$.

DAS28 measures disease activity and is a composite score of Erythrocyte Sedimentation Rate (ESR), number of tender and swollen joints out of 28 joints, and the patient's own estimation of his/her general health on a VAS [21].

### Ethics

All participants provided written and informed consent. The study was conducted according to the Declaration of Helsinki. The regional ethical review board in Gothenburg approved the study (registration number 976–16, November 2016 and supplement T519-17, June 2017).

### Statistics

The power calculation was based on the primary outcome for the ADIRA trial; DAS28. To detect a change in DAS28 of at least 0.6 units, with 90% power and a significance level of 0.05,

38 participants were required. With a sample size of 50 participants, expected dropouts were accounted for.

Because of mostly skewed variables (evaluated through histograms and QQ-plots), results are presented as median (IQR) unless otherwise noticed. Categorical data are expressed as frequency (%).

As main analysis, a linear mixed ANCOVA model was used for all variables, and included participants completing at least one diet period. Treatment (intervention diet or control diet), period (first or second), sequence (intervention diet period or control diet period first) and baseline value of the respective variable were all included as fixed effects in the model, and subject (the participant) was included as a random effect.

Five different types of sensitivity analyses for all outcome variables were performed. Four of them were performed as in previous work (per-protocol [PP] analyses including: 1. only participants who completed the whole study, i.e. both diet periods, 2. only diet periods deemed to have been performed with good compliance, 3. only participants without any changes in DMARD- and/or glucocorticoid use during any of the diet periods, and one intention-to-treat analyses with imputed values for missing data after drop-out) [21], and with the same linear mixed ANCOVA model as for the main analysis. For the fifth sensitivity analysis, a generalized logistic mixed model was used with the outcome variables dichotomized as <median and ≥median for all variables. Treatment, sequence and baseline value of the respective variable were included as fixed effects and subject as random effect in the model.

For all statistical tests, the software IBM SPSS Statistics version 25 (Armonk, NY: IBM Corp.) was used and $P$-values <0.05 were considered significant.

## Results

### Participant characteristics

Sixty-six patients attended the screening visit and 50 were included in the study (Fig 1). With a drop-out rate of 12%, 47 participants completed ≥1 diet period and 44 participants completed the whole study, i.e. completing both intervention and control diet periods. The reasons for drop-out were not related to the study (Fig 1).

Baseline characteristics are presented in Table 1. The majority of the study population were women (77%), the median (IQR) age was 63 (54, 71) years and 49% of the participants had a university degree. Forty-three percent did not work for reasons unknown to the study (e.g., pension because of age or disability). Almost one-third (32%) were obese, and dietary quality at baseline was fair for most participants (79%). The median (IQR) DAS28-ESR at baseline was 3.7 (3.0, 4.7). The majority (57%) had a moderate disease activity (DAS28 3.3–5.1) and 9% had a high disease activity (DAS28 >5.1) [1] (Table 1).

### HrQoL at baseline

Baseline functional ability and other aspects of HrQoL are presented in Table 2. Median (IQR) HAQ-DI was 0.5 (0.1, 1.3), and 68% had a HAQ-DI corresponding to mild physical impairment, 28% a moderate impairment and 4% a severe impairment [36]. Median (IQR) SF-36 PCS and MCS were 42.2 (33.9, 49.0) and 53.9 (44.3, 59.3), respectively. All VAS scales had a median just below 50 mm.

### Adherence and adverse effects

Based on the telephone interview, 91% of the diet periods were completed with good compliance. Details about nutrient intake during the diet periods have been published previously

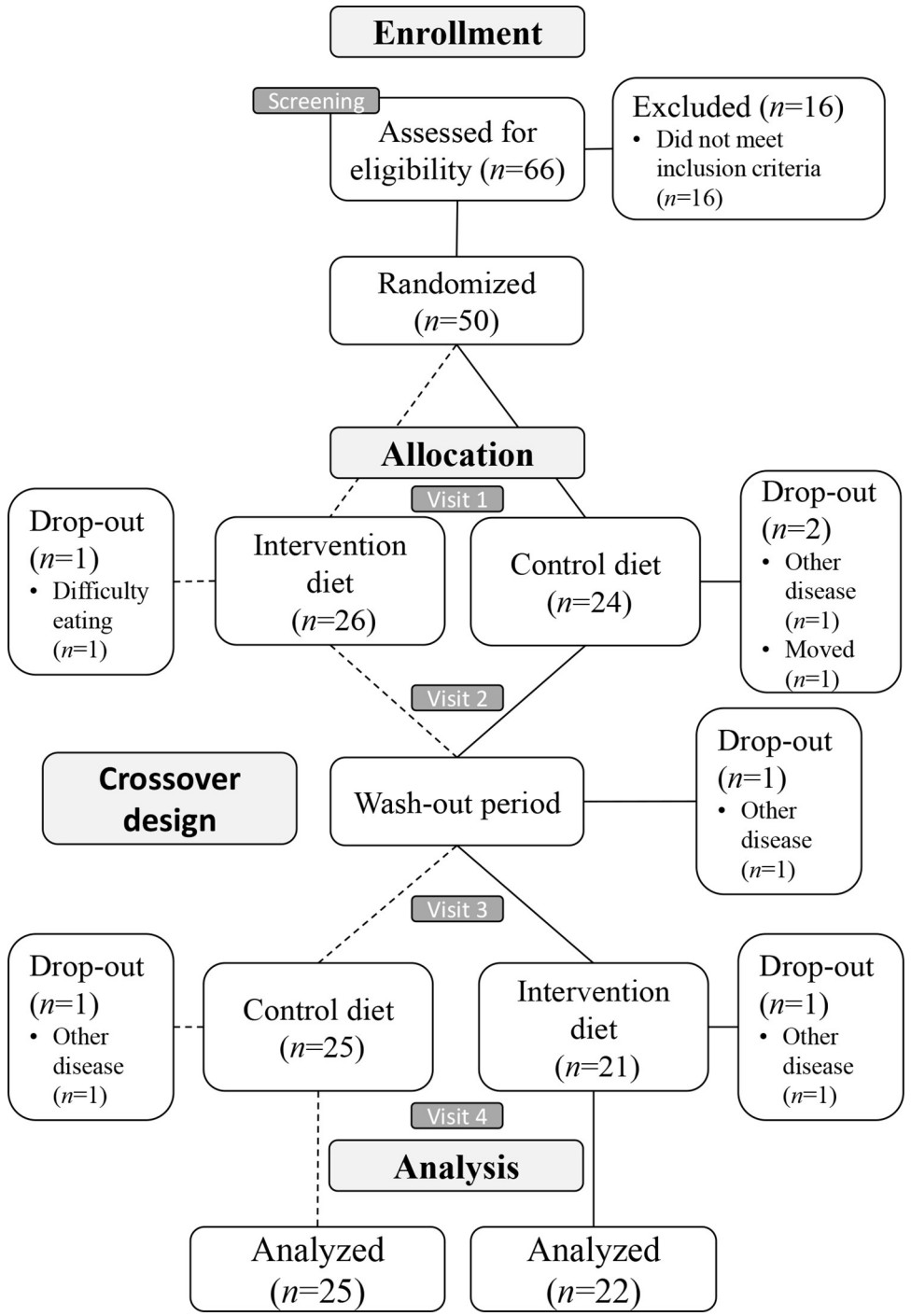

**Fig 1. Flow diagram of the ADIRA trial.**

[21]. Most important, intakes of fiber, polyunsaturated fatty acids (PUFA), eicosapentaenoic acid (EPA) and docosahexaenoic acid (DHA) were significantly higher during intervention diet period (p<0.001), and saturated fatty acid (SFA) intake was significantly higher during control diet period (p<0.001), while energy intake did not differ between the two diet periods (p = 0.675). The latter also was reflected in the absence of significant weight differences

**Table 1. Baseline characteristics of participants in the ADIRA trial.**

| | All (n = 47) | Sequence IC (n = 25) | Sequence CI (n = 22) |
|---|---|---|---|
| | Median (IQR) or n (%) | Median (IQR) or n (%) | Median (IQR) or n (%) |
| **Women** | 36 (76.6) | 20 (80) | 16 (72.7) |
| **Age, years** | 62.8 (53.9, 70.8) | 62.8 (59.3, 70.2) | 64.3 (47.8, 72,4) |
| **Lifestyle factors** | | | |
| Weight, kg | 77.8 (66.9, 85.4) | 75.5 (65.6, 88.9) | 78.0 (70.6, 83.4) |
| BMI, kg/m$^2$ | 26.6 (24.0, 31.8) | 26.9 (23.5, 33.2) | 25.9 (24.1, 30.8) |
| Overweight (BMI 25.0–29.9) | 14 (29.8) | 7 (28) | 7 (31.8) |
| Obese (BMI≥30.0) | 15 (31.9) | 9 (36) | 6 (27.3) |
| Currently smoking | 2 (4.3) | 2 (8) | 0 (0) |
| **Parental birth place** | | | |
| Europe | 44 (93.6) | 23 (92) | 21 (95.5) |
| Africa | 1 (2.1) | 1 (4) | 0 (0) |
| Asia | 2 (4.3) | 1 (4) | 1 (4.5) |
| **Highest educational level** | | | |
| Primary school | 8 (17.0) | 4 (16) | 4 (18.2) |
| 2-y upper secondary school | 9 (19.1) | 4 (16) | 5 (22.7) |
| 3-y upper secondary school | 7 (14.9) | 3 (12) | 4 (18.2) |
| University degree or equal | 23 (48.9) | 14 (56) | 9 (40.9) |
| **Occupational status** | | | |
| Working full time (≥31 h/wk) | 19 (40.4) | 11 (44) | 8 (36.4) |
| Working part-time (≤30 h/wk) | 8 (17.0) | 5 (20) | 3 (13.6) |
| Does not work | 20 (42.6) | 9 (36) | 11 (50) |
| **Dietary intake[a]** | | | |
| Poor dietary quality | 7 (14.9) | 3 (12) | 4 (18.2) |
| Fair dietary quality | 37 (78.7) | 20 (80) | 17 (77.3) |
| High dietary quality | 3 (6.4) | 2 (8) | 1 (4.5) |
| **Disease characteristics** | | | |
| Disease duration, years | 19.2 (10.6, 28.2) | 19.1 (9.7, 28.5) | 19.4 (11.0, 26.5) |
| Seropositive (ACPA and/or RF) | 34 (72.3) | 20 (80) | 14 (63.6) |
| DAS28-ESR | 3.69 (3.03, 4.65) | 3.83 (3.15, 4.60) | 3.34 (2.92, 4.65) |
| Low disease activity (DAS28-ESR≤3.2) | 16 (34.0) | 7 (28) | 9 (40.9) |
| Moderate disease activity (DAS28-ESR 3.3–5.1) | 27 (57.4) | 15 (60) | 12 (54.5) |
| High disease activity (DAS28-ESR>5.1) | 4 (8.5) | 3 (12) | 1 (4.5) |
| **Medication use** | | | |
| *DMARDs* | 42 (89.4) | 23 (92) | 19 (86.4) |
| Anti-TNF | 16 (34.0) | 6 (24) | 10 (45.5) |
| Methotrexate | 31 (66.0) | 15 (60) | 16 (72.7) |
| Sulfalazin | 6 (12.8) | 3 (12) | 3 (13.6) |
| Glucocorticoids | 12 (25.5) | 5 (20) | 7 (31.8) |
| *Analgesics* | | | |
| Non-NSAID analgesics | 13 (27.7) | 5 (20) | 8 (36.4) |
| NSAIDs | 24 (51.1) | 14 (56) | 10 (45.5) |

ACPA, Anti-Citrullinated Protein Antibodies; ADIRA, Anti-inflammatory Diet In Rheumatoid Arthritis; BMI, Body Mass Index; CI, Control–Intervention; DAS28-ESR, Disease Activity Score-28 using Erythrocyte Sedimentation Rate; DMARD, Disease Modifying Anti-Rheumatic Drug; IC, Intervention–Control; NSAID, Non-Steroidal Anti-Inflammatory Drug; RF, Rheumatoid Factor.

Participants completing ≥1 diet period (intervention and/or control) in the randomized crossover ADIRA trial.

[a] Based on a Food Frequency Questionnaire and a dietary quality index adapted from the Swedish National Food Agency [27].

**Table 2. HrQoL at baseline in the ADIRA trial.**

|  | All participants at study start (n = 47) | Pre intervention[a] (n = 46) | Pre control[b] (n = 47) |
|---|---|---|---|
| **HAQ** | 0.5 (0.13, 1.25) | 0.5 (0.13, 1.16) | 0.5 (0.13, 1.25) |
| **SF-36** |  |  |  |
| Physical functioning | 60.0 (40.0, 80.0) | 60.0 (38.8, 81.3) | 65.0 (35.0, 80.0) |
| Role-Physical | 75.0 (43.8, 93.8) | 75.0 (48.4, 90.6) | 75.0 (37.5, 93.8) |
| Bodily Pain | 41.0 (41.0, 62.0) | 51.0 (41.0, 62.0) | 51.0 (41.0, 62.0) |
| General Health | 60.0 (40.0, 77.0) | 60.0 (40.0, 82.0) | 47.0 (40.0, 72.0) |
| Physical Component Summary | 42.2 (33.9, 49.0) | 42.6 (33.8, 50.0) | 40.3 (33.2, 48.0) |
| Vitality | 50.0 (31.3, 68.8) | 53.1 (31.3, 75.0) | 50.0 (37.5, 62.5) |
| Social Functioning | 87.5 (50.0, 100.0) | 100.0 (50.0, 100.0) | 87.5 (50.0, 100.0) |
| Role-Emotional | 91.7 (66.7, 100.0) | 100.0 (75.0, 100.0) | 100.0 (66.7, 100.0) |
| Mental Health | 80.0 (65.0, 90.0) | 82.5 (60.0, 91.3) | 85.0 (70.0, 90.0) |
| Mental Component Summary | 53.9 (44.3, 59.3) | 54.3 (45.8, 60.7) | 54.5 (48.2, 59.0) |
| **VAS Pain (mm)** | 43 (22, 62) | 40 (20, 58) | 36 (19, 61) |
| **VAS Fatigue (mm)** | 48 (22, 61) | 46 (21, 61) | 45 (21, 62) |
| **VAS Morning stiffness (mm)** | 43 (20, 66) | 44 (23, 62) | 43 (21, 64) |
| **Morning stiffness (min)** | 50 (23, 90) | 48 (15, 90) | 55 (25, 90) |

ADIRA, Anti-Inflammatory Diet in Rheumatoid Arthritis; HAQ, Health Assessment Questionnaire; HrQoL, Health-related quality of life; SF-36, 36-item Short Form Survey; VAS, Visual Analogue Scale.

HrQoL at baseline in patients with rheumatoid arthritis participating in the randomized crossover trial ADIRA. Values are presented as median (IQR).

[a] A proposed anti-inflammatory diet.

[b] A diet nutritionally alike Swedish intake.

between the diet periods. In addition, there were no significant weight changes within either diet period, although a trend towards a negligible weight loss during intervention diet period was observed (median [IQR] -0.4 kg [-1.4, 0.6], $p = 0.082$).

Upset stomach was reported as an adverse effect during both diet periods; gas, diarrhea, stomach ache, nausea and heartburn during intervention diet period (13 participants) and constipation, bloating and acid reflux during control diet period (4 participants).

## Effect of intervention on functional ability and other aspects of HrQoL

**HAQ.** There was no significant difference in between intervention diet period and control diet period for HAQ (Tables 3 and S3) in the main analysis. However, although no evidence of difference, when excluding participants with pharmacological treatment changes, the difference between the diet periods, favoring intervention, became much larger (Table 4) (mean: -0.135, 95% CI: -0.270, 0.001, $p = 0.051$, n = 25).

**SF-36v2®.** Although no evidence of difference in the main analysis, physical functioning (SF-36 PF) improved during intervention diet period compared to control diet period, (Tables 3 and S3) (mean:5.392, 95% CI: -0.520, 11.304, $p = 0.073$). In addition, there was a significant improvement during intervention diet period (mean: 5.791, SE: 2.120, 95% CI: 1.576, 10.005). In the sensitivity analyses including only those without pharmacological treatment changes and imputed values for drop outs according to ITT (best case scenario [21]), the difference between intervention and control diet period was significant (mean: 7.898, 95% CI: 0.556, 15.240, $p = 0.036$, n = 25 [Table 4] and mean: 5.709, 95% CI:0.104, 11.315, $p = 0.046$, n = 50 [S1 Table], respectively).

No other aspect of quality of life, using SF-36, exhibited significant differences between the diet periods, but when excluding participants with medication changes, the difference was larger for several of the outcomes, especially the physical domains (Tables 3, 4 and S3).

**Table 3. The effects on HrQoL in the ADIRA trial.**

| | Intervention | | | Control | | | Difference between periods[a] | 95% CIs | p-value |
|---|---|---|---|---|---|---|---|---|---|
| | Mean change | SE | 95% CI | Mean change | SE | 95% CI | | | |
| **HAQ[b]** | -0.058 | 0.049 | -0.155, 0.039 | -0.017 | 0.048 | -0.113, 0.078 | -0.041 | -0.162, 0.081 | 0.503 |
| **SF-36[b]** | | | | | | | | | |
| Physical Functioning | 5.791 | 2.120 | 1.576, 10.005 | 0.399 | 2.086 | -3.748, 4.545 | 5.392 | -0.520, 11.304 | 0.073 |
| Role-Physical | 5.052 | 3.020 | -0.954, 11.057 | 4.450 | 2.972 | -1.459, 10.358 | 0.602 | -7.586, 8.790 | 0.883 |
| Bodily Pain | 1.212 | 2.141 | -3.044, 5.468 | -0.287 | 2.107 | -4.475, 3.900 | 1.499 | -4.472, 7.471 | 0.619 |
| General Health | -2.275 | 1.855 | -5.962, 1.413 | 0.887 | 1.826 | -2.744, 4.517 | -3.161 | -8.208, 1.885 | 0.212 |
| Physical Component Summary | 0.758 | 0.831 | -0.894, 2.410 | 0.739 | 0.818 | -0.886, 2.364 | 0.019 | -2.300, 2.338 | 0.987 |
| Vitality | -0.062 | 2.671 | -5.371, 5.248 | 2.911 | 2.628 | -2.314, 8.136 | -2.973 | -10.408, 4.463 | 0.424 |
| Social Functioning | 0.711 | 2.813 | -4.881, 6.303 | 1.293 | 2.768 | -4.209, 6.795 | -0.582 | -8.427, 7.263 | 0.883 |
| Role-Emotional | 5.181 | 2.725 | -0.248, 10.611 | 1.125 | 2.688 | -4.234, 6.483 | 4.057 | -1.920, 10.033 | 0.178 |
| Mental Health | 1.825 | 2.054 | -2.259, 5.908 | 0.419 | 2.019 | -3.596, 4.434 | 1.406 | -4.094, 6.906 | 0.608 |
| Mental Component Summary | 0.781 | 1.134 | -1.474, 3.036 | 0.437 | 1.116 | -1.782, 2.656 | 0.344 | -2.630, 3.318 | 0.816 |
| **VAS Pain (mm)[b]** | 0.210 | 3.468 | -6.683, 7.104 | 2.676 | 3.412 | -4.107, 9.460 | -2.466 | -12.145, 7.213 | 0.610 |
| **VAS Fatigue (mm)[b]** | -2.588 | 3.442 | -9.432, 4.255 | -0.035 | 3.387 | -6.768, 6.999 | -2.554 | -12.073, 6.966 | 0.591 |
| **VAS Morning stiffness (mm)[b]** | -1.200 | 2.828 | -6.822, 4.421 | -2.918 | 2.782 | -8.450, 2.613 | 1.718 | -6.174. 9.610 | 0.666 |
| **Morning stiffness (min)[b]** | -0.354 | 5.010 | -10.314, 9.607 | -4.102 | 4.930 | -13.903, 5.699 | 3.748 | -10.233, 17.730 | 0.595 |

ADIRA, Anti-inflammatory Diet In Rheumatoid Arthritis; HAQ, Health Assessment Questionnaire; HrQoL, Health-related Quality of Life; SE, Standard error; SF-36, 36-item Short Form Health Survey; VAS, Visual Analogue Scale.

Modelled estimates of differences in effects of a proposed anti-inflammatory diet (intervention) compared to a diet nutritionally alike usual Swedish intake (control) in patients with rheumatoid arthritis in the randomized controlled crossover trial ADIRA. Participants completing at least one diet period (n = 47).

[a] Differences at the end of diet periods (Intervention–Control).

[b] Linear mixed model with period, treatment, sequence and baseline value as fixed effects and subject as random effect.

**Pain, fatigue and morning stiffness.** For VAS pain, there were no significant differences between the diet periods in any of the analyses (Tables 3, 4, S1 and S2).

The main analysis of fatigue did not result in any significant difference (Tables 3 and S3) but, although no evidence of difference, when participants with changes in the pharmacological treatment were excluded, the difference in fatigue after intervention diet period compared to control diet period was much larger (Table 4) (mean: -9.068, 95% CI: -19.713, 1.577, $p = 0.091$, n = 25).

No significant differences were observed for VAS morning stiffness or duration of morning stiffness (Tables 3, 4, S1 and S2).

## Discussion

In the ADIRA trial, the proposed anti-inflammatory diet induced some improvements in HrQoL in patients with RA. Physical functioning (SF-36 PF) improved significantly during intervention period and, although no evidence of difference, this improvement was large also compared to control diet period. When participants with changes in their pharmacological treatment were excluded from the main analyses, the differences between the anti-inflammatory diet and the control diet were larger for several of the HrQoL outcomes.

The term *Health-related Quality of Life* can be defined as a sub-group of the broader term *Quality of Life*; however, a clear and generally accepted definition of these terms is lacking

**Table 4. The effects on HrQoL in participants without medication changes in the ADIRA trial.**

| | Intervention | | | Control | | | Difference between periods[a] | 95% CIs | p-value |
|---|---|---|---|---|---|---|---|---|---|
| | Mean change | SE | 95% CI | Mean change | SE | 95% CI | | | |
| **HAQ[b]** | -0.070 | 0.047 | -0.166, 0.026 | 0.065 | 0.047 | -0.031, 0.160 | -0.135 | -0.270, 0.001 | 0.051 |
| **SF-36[b]** | | | | | | | | | |
| Physical Functioning | 7.310 | 2.578 | 2.119, 12.502 | -0.588 | 2.578 | -5.779, 4.604 | 7.898 | 0.556, 15.240 | 0.036 |
| Role-Physical | 5.997 | 3.187 | -0.422, 12.416 | 2.291 | 3.187 | -4.127, 8.710 | 3.706 | -5.371, 12.783 | 0.415 |
| Bodily Pain | 1.639 | 2.639 | -3.677, 6.955 | -0.909 | 2.641 | -6.229, 4.411 | 2.548 | -4.975, 10.070 | 0.499 |
| General Health | -1.748 | 2.650 | -7.090, 3.594 | -0.294 | 2.649 | -5.634, 5.046 | -1.454 | -9.093, 6.185 | 0.690 |
| Physical Component Summary | 1.481 | 0.983 | -0.499, 3.461 | 0.115 | 0.984 | -1.866, 2.096 | 1.366 | -1.436, 4,169 | 0.331 |
| Vitality | -2.761 | 3.125 | -9.171, 3.649 | 0.620 | 3.124 | -5.789, 7.029 | -3.381 | -9.350, 2.588 | 0.248 |
| Social Functioning | 3.119 | 2.936 | -2.795, 9.034 | -0.852 | 2.937 | -6.768, 5.064 | 3.971 | -4.411, 12.353 | 0.345 |
| Role-Emotional | 2.700 | 3.750 | -4.998, 10.399 | 0.976 | 3.749 | -6.721, 8.674 | 1.724 | -5.918, 9.366 | 0.635 |
| Mental Health | 0.990 | 2.534 | -4.127, 6.107 | -0.140 | 2.535 | -5.259, 4.979 | 1.130 | -6.081, 8.342 | 0.738 |
| Mental Component Summary | -0.149 | 1.385 | -2.949, 2.650 | 0.083 | 1.385 | -2.717, 2.884 | -0.233 | -3.979, 3.513 | 0.895 |
| **VAS Pain mm[b]** | 0.004 | 4.672 | -9.409, 9.416 | 0.678 | 4.673 | -8.736, 10.092 | -0.675 | -13.873, 12.524 | 0.917 |
| **VAS Fatigue mm[b]** | -4.514 | 4.580 | -13.777, 4.750 | 4.554 | 4.580 | -4.709, 13.818 | -9.068 | -19.713, 1.577 | 0.091 |
| **VAS Morning stiffness mm[b]** | 0.359 | 3.961 | -7.618, 8.336 | -1.769 | 3.965 | -9.754, 6.217 | 2.127 | -9.160, 13.414 | 0.706 |
| **Morning stiffness min[b]** | 1.268 | 5.120 | -9.056, 11.592 | 4.668 | 5.113 | -5.643, 14.979 | -3.400 | -17.452, 10.652 | 0.617 |

ADIRA, Anti-inflammatory Diet In Rheumatoid Arthritis; HAQ, Health Assessment Questionnaire; HrQoL, Health-related Quality of Life; SE, Standard error; SF-36, 36-item Short Form Health Survey; VAS, Visual Analogue Scale. Modelled estimates of differences in effects of a proposed anti-inflammatory diet (intervention) compared to a diet nutritionally alike usual Swedish intake (control) in participants without changes in Disease-Modifying Anti-Rheumatic Drugs and glucocorticoids in the randomized controlled crossover trial ADIRA (n = 25).

[a] Differences at the end of diet periods (Intervention–Control).

[b] Linear mixed model with period, treatment, sequence and baseline value as fixed effects and subject as random effect.

[37]. In addition, one may argue that only patients themselves can define what entails QoL. Nevertheless, certain outcomes are by many regarded as parts of QoL; e.g. physical functioning, level of pain and social life. Although these may be assessed objectively by physical or mental tests, or from relatives or physicians, the patient's own estimation as well as the use of valid instruments are crucial to capture QoL.

Commonly, HrQoL questionnaires collect answers that represent categorical data on an ordinal scale, e.g., *Without any difficulties*, *Unable to do*, *Most of the time*, *None of the time*. Often, these answers are converted to numbers and summed up to an overall score that is statistically handled as a continuous, normally distributed variable. Walters et al [38] point out several problems with this approach, e.g., the lack of constant variance and that changes over time may depend on the initial value–participants starting with low scores have a chance to larger improvements than those starting with high scores if the scale is capped in the upper end. In the present study, we also chose to treat our HrQoL outcomes as continuous normally distributed variables. This allows us to compare our results with those of others. In addition, for the main analyses we confirmed that the residuals of our regression analyses were normally distributed for all outcomes by evaluating QQ-plots and histograms. We also performed sensitivity analyses using a generalized logistic mixed model with the outcome variables

dichotomized and hence treated as nominal variables. In this, much information in the original data was lost and not surprisingly none of those analyses was significant. However, the sensitivity analyses provided results that were in the same direction as our main analyses for the majority of the outcomes, indicating that the linear mixed ANCOVA model was adequate to use for our analyses.

Few of our analyses were statistically significant. One reason for this may be that our power calculation and study length were based on the ADIRA trial's primary outcome; DAS28. Perhaps 50 participants were too few to detect statistically significant differences in HrQoL outcomes. Also, ten weeks may have been insufficient time to experience improvements in self-perceived HrQoL, from dietary change. In addition, adverse effects, supposedly affecting quality of life negatively, were reported more often during the intervention period. The higher fiber intake compared to their baseline intake [21] could explain these adverse effects, and this problem probably would have decreased had the trial continued for longer. Second, our sample of patients with RA may have been too healthy to experience improvements in SF-36 and HAQ in the range of these instruments with optimal performance [39]. Third, since several of the outcomes exhibited larger improvements when participants with changes in pharmacological treatment were excluded, enforcing restrictions in medication changes likely could have resulted in significant differences. However, this would probably have led to a higher drop-out rate.

We evaluated multiple aspects of HrQoL using several instruments. We did not adjust the predetermined significance level for number of tests but instead urge for caution when interpreting our results due to the higher risk of false positive results. In addition, very few of our analyses had a p-value below the predetermined significance level of 0.05.

Still, statistical significance is not all that matters to patients; equally important is whether the obtained effect is clinically relevant [40]. The Minimal/Minimum (Clinical) Important Difference (M[C]ID) describes what is a meaningful and clinically relevant change, and reporting this may help clinicians and patients interpret research results [41]. In ADIRA, the mean difference between the diet periods in both the main analysis and several of the sensitivity analyses of SF-36 PF reached previously proposed MCIDs [42]. In addition, the mean change during the intervention period for SF-36 PF as well as RE also reached an effect meaningful in clinical setting. Some of these analyses had a p-value of <0.05. Even for HAQ, the mean difference between the diet periods in the analysis including those without medication changes can be considered clinically relevant [29,41] although this was not statistically significant. In addition, in the same sensitivity analysis, the difference between the diet periods for VAS Fatigue almost reached the proposed MCID [43] in that analysis as well. However, one should be aware of that calculated MCIDs and MIDs differ between reports because of different methods used to estimate the MCIDs and MIDs, different study populations and varying disease severity [44]. Finally, one could also argue that all improvements in HrQoL are of importance to the individual, and therefore are of clinical relevance.

Clinical trials investigating the effects of a whole diet on HrQoL in RA are scarce. One trial evaluated a year of fasting followed by a gluten-free vegan diet and thereafter a lacto-vegetarian diet, and this resulted in a higher functional ability and less pain as well as shorter duration of morning stiffness [45]. In contrast, another trial used an uncooked vegan diet rich in lactobacilli, but did not obtain any differences in these same outcomes [46]. The ADIRA diet is comparable to a Mediterranean diet but with the addition of probiotics, and there have been a few previous interventions with Mediterranean diets [4,5,47]. Among these, the study by Sköldstam et al is most similar to the ADIRA trial [4]. In contrast to our results, Sköldstam et al observed significant improvements in HAQ, pain and vitality (SF-36 VT), while we saw significantly improved SF-36 PF during intervention period. The disparate results could be due to

between study differences in patient well-being at baseline, DMARD use during study time and statistical approaches. In another trial with the Mediterranean diet, McKellar et al's study [5], differences in HAQ, pain and duration of morning stiffness between the Mediterranean diet and the control diet ("healthy eating") were all significant after three and/or six months, favoring the Mediterranean diet. However, this was a non-randomized pilot study conducted in areas of social deprivation and therefore different from the ADIRA trial. Finally, the third study using Mediterranean diet by Abendroth et al [47] resulted in some improvements in HrQoL. Still, this study also was non-randomized, the control group was fasting, and the participants received multimodal medicine treatment consisting of physical activity, a programme for stress reduction, hydrotherapy etc.; thus making it difficult to compare to our study.

## Limitations

This study has some limitations that should be addressed. First, all outcomes here reported are subjective and hence vulnerable to preconceptions among participants. Unfortunately, it is notoriously difficult to double-blind a dietary intervention. We did however communicate to the participants that we were evaluating effects from two diets and, considering comments and questions received, it seems that we succeeded in blinding many participants. However, some participants likely held preconceptions about a healthy diet and this could possibly have affected their subjective outcomes. Still, if objective outcome measures had been used important aspects of HrQoL may have been missed—objective measures do not capture all important aspects of the RA disease [39]. In the recommendations on how disease activity should be reported in clinical trials formed by the European Alliance of Associations for Rheumatology (EULAR) and American College of Rheumatology (ACR), ACR response, which includes both patient-reported pain and disability [39], is listed [48]. In addition, they recommend fatigue to be assessed in clinical trials.

Second, there are several aspects that may be included in the definition of HrQoL which we did not evaluate; e.g. quality of sleep, sexual functioning, self-esteem. However, to our knowledge no evidence exists for diet to affect these outcomes in patients with RA. Finally, as discussed earlier, our power calculation and study length were based on the ADIRA trial's primary outcome; DAS28, and for ethical reasons we did not restrict changes in pharmacological treatment during the study period. Both these factors probably affected our ability to detect significant results.

## Strengths

This study has several strengths. The randomized crossover design minimized the risk for confounding factors and by using the SRQ, we ascertained that all possibly eligible participants in this area of Sweden were invited to the study. Furthermore, people do not consume nutrients or food components separately; they consume a diet. Therefore, to investigate a whole diet including several of the food components that seem to affect RA symptoms ensures that possible potentiating effects caused by their interaction with each other are captured. In addition, providing participants with menus, recipes, food ingredients and ready-to-eat meals through home delivery increases the likelihood of good compliance. This was also something we, through an interview and food records, could conclude eventually; the vast majority of the participants had a high compliance to both diets. Further, recently published, changes in plasma fatty acids indicated good compliance at a group level [49]. Validated and widely used instruments, only PROs and both disease-specific instruments more responsive to change (HAQ), and generic instruments which are able to detect unexpected effects and make it possible to compare results to other populations [42], were used, giving a fair picture of the patients' self-

perceived well-being and HrQoL. Finally, QoL-studies often suffer from many missing values [38]. This was not the case in our study, which indicates a proper choice of instruments and high level of participation among the patients.

## Conclusions

In this study evaluating effects on HrQoL of a proposed anti-inflammatory diet, main analyses did not show significant improvements in any outcome. However, during the intervention diet period and in the analysis where participants with changes in anti-rheumatic medication were excluded, physical functioning improved significantly. Larger studies with restrictions in medication changes conducted in study populations more affected by the disease are needed to confirm our results and to possibly detect significant changes in other HrQoL-outcomes as well.

## Supporting information

**S1 Table. The effects on HrQoL in the ADIRA trial (sensitivity analyses).** Modelled estimates of differences in effects of a proposed anti-inflammatory diet (intervention) compared to a diet nutritionally alike usual Swedish intake (control) in patients with rheumatoid arthritis in the randomized controlled crossover trial ADIRA, with imputed values for missing data due to drop-out.
(PDF)

**S2 Table. The effects on HrQoL in the ADIRA trial (sensitivity analyses).** Modelled estimates of differences in effects of a proposed anti-inflammatory diet (intervention) compared to a diet nutritionally alike usual Swedish intake (control) in sensitivity analyses in the randomized controlled crossover trial ADIRA.
(PDF)

**S3 Table. The effects on HrQoL in the ADIRA trial.** Modelled estimates of differences in effects of a proposed anti-inflammatory diet (intervention) compared to a diet nutritionally alike usual Swedish intake (control) in patients with rheumatoid arthritis in the randomized controlled crossover trial ADIRA.
(PDF)

**S1 File. CONSORT checklist.**
(PDF)

**S2 File. Study protocol.**
(PDF)

## Acknowledgments

We want to thank all participants for their dedicated participation in the ADIRA trial.

## Author Contributions

**Conceptualization:** Linnea Bärebring, Inger Gjertsson, Helen M. Lindqvist, Anna Winkvist.

**Data curation:** Linnea Bärebring, Helen M. Lindqvist.

**Formal analysis:** Anna Turesson Wadell.

**Funding acquisition:** Linnea Bärebring, Anna Winkvist.

**Investigation:** Anna Turesson Wadell, Linnea Bärebring, Erik Hulander, Helen M. Lindqvist.

**Methodology:** Linnea Bärebring, Inger Gjertsson, Lars Hagberg, Helen M. Lindqvist, Anna Winkvist.

**Project administration:** Linnea Bärebring, Helen M. Lindqvist, Anna Winkvist.

**Resources:** Inger Gjertsson, Anna Winkvist.

**Supervision:** Anna Winkvist.

**Writing – original draft:** Anna Turesson Wadell.

**Writing – review & editing:** Anna Turesson Wadell, Linnea Bärebring, Erik Hulander, Inger Gjertsson, Lars Hagberg, Helen M. Lindqvist, Anna Winkvist.

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
