## [Decision Letter · Decision Letter 0]

11 Jun 2021

PONE-D-20-30996

Effects on health-related quality of life in the randomized, controlled crossover trial ADIRA (Anti-inflammatory Diet In Rheumatoid Arthritis)

PLOS ONE

Dear Dr. Winkvist,

Thank you for submitting your manuscript to PLOS ONE. After careful consideration, we feel that it has merit but does not fully meet PLOS ONE’s publication criteria as it currently stands. Therefore, we invite you to submit a revised version of the manuscript that addresses the points raised during the review process.

We look forward to receiving your revised manuscript.

Kind regards,

Yuanyuan Wang, PhD

Academic Editor

PLOS ONE

Journal Requirements:

Additional Editor Comments:

The reviewers have raised some methodological issues that the authors will need to consider and address, such as sample size, confounding, and adherence to intervention.

Reviewers' comments:

Reviewer's Responses to Questions

**Comments to the Author**

1. Is the manuscript technically sound, and do the data support the conclusions?

Reviewer #1: Yes

Reviewer #2: Partly

Reviewer #3: No

2. Has the statistical analysis been performed appropriately and rigorously? 

Reviewer #1: Yes

Reviewer #2: Yes

Reviewer #3: I Don't Know

3. Have the authors made all data underlying the findings in their manuscript fully available?

Reviewer #1: No

Reviewer #2: Yes

Reviewer #3: No

4. Is the manuscript presented in an intelligible fashion and written in standard English?

Reviewer #1: Yes

Reviewer #2: No

Reviewer #3: Yes

5. Review Comments to the Author

Reviewer #1: Abstract

- line 5: "... pharmacological treatment does not always resolve these factors."

- line 18: there's nothing in the reported result that suggests a trend towards better physical functioning; this should be removed.

- line 20 to 23: are the results presented mean scores at a particular time or changes in mean scores? If so, the results should explain this more clearly.

- what is the reason for excluding participants with medication changes - is this a per-protocol analysis? If so this should be stated more explicitly.

- line 25: the difference and its confidence interval (and p-value if desired) should be reported.

Methods

- please re-consider whether it is really necessary to go through the procedure described for identifying confounders to adjust for. If the sequence allocation was randomised, then this procedure is not necessary, and adjustment for age and sex only is probably sufficient. Furthermore most of what is described in lines 222 to 230 are actually results.

Results

- Table 1 could be more informative if you had separate columns for those randomised to receive control diet first and those randomised to receive intervention diet first (do not conduct statistical tests comparing the groups if you do this)

- please indicate the differences (including units) and 95% confidence intervals for the intakes described in lines 288 to 290

- please avoid phrases such as "a trend towards /negligible/significant..." e.g. in line 18, line 292, unless you have actually explored the statistical evidence for the trend you describe. Also avoid phrases such as "close to significant" e.g. line 303 - a result is either statistically significant (more appropriately, there is evidence for a difference) or not.

- line 300: this sentence should be describing a difference in some outcome, not a difference in analysis as it seems to imply; please re-word this appropriately, e.g. "There was no evidence of a difference in HAQ between the intervention diet period and control diet period."

- Table 3 should report the mean changes and the standard errors, not confidence intervals, of the mean changes in the second and third columns (titled 'intervention' and 'control')

- The structure of Table 4 should match that of Table 3, i.e. the same information should be provided.

Discussion

- line 365 and 366 and elsewhere: avoid phrases such as "did not reach significance", "reached significance" (line 362/3) for the reasons explained above, and statements suggesting a trend towards some outcome, as this has not been formally tested.

Reviewer #2: This is an interesting paper, however, some change should be made:

I am concerned about the energy prescription (1100 kcal/day) because it should be a personalized one, the same for protein prescription this should be mentioned in the discussion.

The authors did not mention if they evaluated physical activity and, diet consumed the 2 days a week that patients did not receive the anti-inflammatory diet.

In table 1 include the weight and comorbidities of studied patients

The authors should include the following references in the discussion:

Ideal food pyramid for patients with rheumatoid arthritis: a narrative review. Clinical Nutrition 2021;40:661-689.

Effect of a Dynamic Exercise Program in Combination With Mediterranean Diet on Quality of Life in Women With Rheumatoid Arthritis. J Clin Rheumatol 2020;26(7S Suppl 2):S116-S122. doi: 10.1097/RHU.0000000000001064.

In the discussion HrQoL and statistical significance are mentioned could be abbreviated

Reviewer #3: Thank you to the authors for this interesting study. I appreciate that studies investigating dietary interventions can be particularly challenging and often with unavoidable limitations. I include some thoughts below:

Major

-There are clearly multiple confounders and I am not clear as to how the authors took account of these in their analyses. Aside from sociodemographic factors and other patient and disease characteristics, how about the fact that pharmacological treatment was allowed to vary – how could this have affected the results?

-The sample size was small and potentially subject to selection bias. Could these be reasons for the results seen and the lack of significance in at least some of the findings?

-The study duration was less than 3 months, which is an important limitation. Have the authors continued with the study, do they plan to present longer-term results and is there any plan for more objective outcomes to be studied?

-How certain can we be about patients adhering to the specific dietary interventions? Do the authors think that a phone call and the patients’ ‘word’ was enough, could this have been an issue?

-Baseline BMI in these patients was 26.6? What do the authors think about this, importantly in relation to the outcomes study, which are all subjective?

-Could the authors elaborate in the discussion regarding the observations after exclusion of pharmacological treatments from the analyses?

Minor

-The text throughout could be more concise to my opinion. The introduction for example and discussion are long and could be tightened up, also for messages to be more clearly relayed to the reader.

-The EULAR acronym should be updated to the correct full name (European Alliance of Associations for Rheumatology).

6. PLOS authors have the option to publish the peer review history of their article (what does this mean?). If published, this will include your full peer review and any attached files.

Reviewer #1: No

Reviewer #2: No

Reviewer #3: No

---

## [Author Response · Author response to Decision Letter 0]

16 Jul 2021

REBUTTAL LETTER

MS ID#: PONE-D-20-30996

Title: Effects on health-related quality of life in the randomized, controlled crossover trial ADIRA (Anti-inflammatory Diet In Rheumatoid Arthritis)

Comments from the Editors and Reviewers:

Journal Requirements:

Response: We have checked and made minor revisions accordingly.

Response: Data cannot be made freely available as they are subject to secrecy in accordance with the Swedish Public Access to Information and Secrecy Act [Offentlighets- och sekretesslagen, OSL, 2009:400], but can be made available to researchers upon request (subject to a review of secrecy). Requests for data should be made to Anna Winkvist, anna.winkvist@nutrition.gu.se. Requests could also be addressed to the head of the Department of Internal Medicine and Clinical Nutrition at the University of Gothenburg, Sweden, Jorgen.isgaard@medic.gu.se

This explanation and contact information for data access have now been added to the revised cover letter.

Response: This has now been corrected and we apologize for the inconsistent information. 

Additional Editor Comments:

The reviewers have raised some methodological issues that the authors will need to consider and address, such as sample size, confounding, and adherence to intervention.

Review Comments to the Author

Reviewer #1:

Thanks for taking the time to review our manuscript. Your points and suggestions have definitely helped us to improve the manuscript. Please find our responses to your comments below each comment.

 Abstract

- line 5: "... pharmacological treatment does not always resolve these factors."

Response: This has been corrected

- line 18: there's nothing in the reported result that suggests a trend towards better physical functioning; this should be removed.

Response: This has been removed.

- line 20 to 23: are the results presented mean scores at a particular time or changes in mean scores? If so, the results should explain this more clearly.

Response: This has been clarified in line 20.

- what is the reason for excluding participants with medication changes - is this a per-protocol analysis? If so this should be stated more explicitly.

Response: We chose to describe this in lines 232-238 in the Methods section since the abstract should be short and concise. The main analysis is by intention to treat. Thereafter we performed 5 sensitivity analyses, all of which are per protocol. The analysis of only those without medication changes is one of these per protocol sensitivity analyses. 

- line 25: the difference and its confidence interval (and p-value if desired) should be reported.

Response: This has been corrected.

Methods

- please re-consider whether it is really necessary to go through the procedure described for identifying confounders to adjust for. If the sequence allocation was randomised, then this procedure is not necessary, and adjustment for age and sex only is probably sufficient. 

Furthermore most of what is described in lines 222 to 230 are actually results.

Response: We agree and have removed unnecessary information. It is correct that the randomization procedure likely removed confounding effects of covariates, but for safety we chose to evaluate some common potential confounding factors. 

Results

- Table 1 could be more informative if you had separate columns for those randomised to receive control diet first and those randomised to receive intervention diet first (do not conduct statistical tests comparing the groups if you do this)

Response: This has been added.

- please indicate the differences (including units) and 95% confidence intervals for the intakes described in lines 288 to 290

Response: To keep the results section shorter and since the nutrient intake during the diet periods has already been published, we chose to refer to a previous publication instead, and only presented the most important information in this manuscript.

- please avoid phrases such as "a trend towards /negligible/significant..." e.g. in line 18, line 292, unless you have actually explored the statistical evidence for the trend you describe. Also avoid phrases such as "close to significant" e.g. line 303 - a result is either statistically significant (more appropriately, there is evidence for a difference) or not.

Response: The p-values are 0.082, 0.073 and 0.089, hence not statistically significant. However, with such small p-values, in light of our small sample size, it would be unwise to draw the conclusion of no change at all. Therefore, we want to keep these statements and indicate that something may be going on that could be evaluated with a larger sample size/stronger study design in the future. 

- line 300: this sentence should be describing a difference in some outcome, not a difference in analysis as it seems to imply; please re-word this appropriately, e.g. "There was no evidence of a difference in HAQ between the intervention diet period and control diet period."

Response: Thanks for pointing this out! This has now been corrected.

- Table 3 should report the mean changes and the errors, not confidence intervals, of the mean changes in the second and third columns (titled 'intervention' and 'control')

Response: Whether SE or CI is preferred seems to differ among statisticians. This table structure was the one our statistician suggested. Therefore, we wish to keep the confidence intervals in our main table in the manuscript. However, SEs can be found in Supplemental Table 3. 

- The structure of Table 4 should match that of Table 3, i.e. the same information should be provided.

Response: Table 4 now matches the structure of Table 3.

Discussion

- line 365 and 366 and elsewhere: avoid phrases such as "did not reach significance", "reached significance" (line 362/3) for the reasons explained above, and statements suggesting a trend towards some outcome, as this has not been formally tested.

Response: We have rephrased line 358-359, 365, 388-389, 392, 409 and 422. See previous comment above on our thoughts on trends though.

Reviewer #2: This is an interesting paper, however, some change should be made:

Thank you very much! Our responses to your comments can be found below each comment.

I am concerned about the energy prescription (1100 kcal/day) because it should be a personalized one, the same for protein prescription this should be mentioned in the discussion.

Response: We understand your concern. However, the food corresponding to 1100 kcal/day was the food we provided, but they consumed additionally food as much as they wanted and/or needed (with instructions to keep a stable weight). Thus, the energy and protein “prescriptions” actually were personally ones. We do however agree that this can be more clear in the method section and have therefore added this to the description of the dietary intervention in lines 102-104.

The authors did not mention if they evaluated physical activity and, diet consumed the 2 days a week that patients did not receive the anti-inflammatory diet.

Response: That is a good point! We asked participants about physical activity at baseline but not after each diet period since they were instructed to keep the same activity throughout the study periods. Once again, being a crossover study the comparison is not between different participants. The days they did not eat food provided by us they were still instructed to eat the same type of foods as those provided by us. We evaluated the nutrient intake during each diet period using 3-d food records (line 285-290) and for some participants one or two days of these could possibly have been such days. Most important is that we could see differences between polyunsaturated fatty acids, EPA, DHA, saturated fatty acids and fiber – indicating that they overall were compliant to the diets. We have recently published results on these changes in plasma fatty acids during the diet periods, which, at a group level, indicated a good compliance to the diets. Those results were not published when this manuscript was submitted, but have now been added to the discussion (lines 500-502) as well as the reference list. Further, we also plan to analyze additional diet biomarkers to be able to evaluate adherence objectively.

In table 1 include the weight and comorbidities of studied patients

Response: Weight has been added. In case of serious disease, the patient was not included in the study at all (lines 89-91).

The authors should include the following references in the discussion:

Ideal food pyramid for patients with rheumatoid arthritis: a narrative review. Clinical Nutrition 2021;40:661-689.

Effect of a Dynamic Exercise Program in Combination With Mediterranean Diet on Quality of Life in Women With Rheumatoid Arthritis. J Clin Rheumatol 2020;26(7S Suppl 2):S116-S122. doi: 10.1097/RHU.0000000000001064.

Response: Thanks for your suggestions on articles to use as references! We have added the article by Rondanelli et al as a reference (lines 41-42). However, since our intervention does not include physical activity, we find it less useful to compare our results with the ones in the article by García-Morales et al.

In the discussion HrQoL and statistical significance are mentioned could be abbreviated

Response: We have now tightened up several parts of the discussion. 

Reviewer #3: Thank you to the authors for this interesting study. I appreciate that studies investigating dietary interventions can be particularly challenging and often with unavoidable limitations. I include some thoughts below:

Thank you very much for those kind words and your thoughts. See our responses below each comment.

Major

-There are clearly multiple confounders and I am not clear as to how the authors took account of these in their analyses. Aside from sociodemographic factors and other patient and disease characteristics, how about the fact that pharmacological treatment was allowed to vary – how could this have affected the results?

Response: We agree that the varying pharmacological treatment is of concern. There were some other confounders which we adjusted for and this is described in the Methods section. However, since this is a crossover study, each participant is his/her own control and therefore confounding effects are not such a severe issue as in parallel intervention studies. Still, there were some changes in medication among participants during the study, and to explore effects of this we chose to exclude all participants with such changes in a per protocol sensitivity analysis. Although the population became much smaller, we could see important changes in study effects. This means that changes in pharmacological treatment affected the results and we do address this issue in lines 402-405, 446-448 and 487-489.

-The sample size was small and potentially subject to selection bias. Could these be reasons for the results seen and the lack of significance in at least some of the findings?

Response: We agree, the sample size is small. Yes, there could possibly be a selection bias. Probably many of the participants had a certain interest in food and health, maybe a larger proportion than in the whole population with RA. In addition, for many to even consider it possible to participate in this kind of studies, they cannot be too ill. The issue of having a somewhat healthier study population compared to the general RA population is addressed in lines 400-402.

-The study duration was less than 3 months, which is an important limitation. Have the authors continued with the study, do they plan to present longer-term results and is there any plan for more objective outcomes to be studied?

Response: This is a good point! We agree that ten weeks may be too short to see changes in these subjective QoL outcomes (discussed in lines 395-400) but the main outcome for ADIRA was DAS28; thus study duration was planned mainly for that and not for the QoL outcomes. However, we have no plans to continue this study to explore whether participants have continued with the diets and if so, how they experience their life quality now.

-How certain can we be about patients adhering to the specific dietary interventions? Do the authors think that a phone call and the patients’ ‘word’ was enough, could this have been an issue?

Response: The lack of compliance can absolutely be a problem in dietary interventions and the discussion on how to examine this in a proper way is always present. In our study, except for the phone call, participants also performed a 3-d food record during the diet periods. Food records are the golden standard to measure dietary intake and in lines 287-292 we present some results from these, which at least indicate adherence to the diets. In addition, we have recently published results on changes in plasma fatty acids during the diet periods, which, at a group level, indicated a good compliance to the diets. Those results were not published when this manuscript was submitted, but have now been added to the discussion (lines 500-502) as well as the reference list. Further, we also plan to analyze additional diet biomarkers to be able to evaluate adherence objectively.

-Baseline BMI in these patients was 26.6? What do the authors think about this, importantly in relation to the outcomes study, which are all subjective?

Response: Since this is a crossover study (each participant is his/her own control) this should not affect the difference in outcomes between diet periods.

-Could the authors elaborate in the discussion regarding the observations after exclusion of pharmacological treatments from the analyses?

Response: We agree that this is an important part of the discussion. However, although there are several results and parts of the study design that could be discussed in more detail, we had to choose what parts we wanted to discuss more thoroughly to avoid an even longer discussion. We have already discussed briefly what results we may have obtained with stricter control of the pharmacological treatment. On the other hand, we were unable to restrict the use of medication due to ethical reasons and even if; this would probably have led to a higher drop-out rate (which is now added to line 405).

Minor

-The text throughout could be more concise to my opinion. The introduction for example and discussion are long and could be tightened up, also for messages to be more clearly relayed to the reader.

Response: We agree and have now tightened up these sections.

-The EULAR acronym should be updated to the correct full name (European Alliance of Associations for Rheumatology).

Response: Thank you very much for the information about the new name. This has been corrected.

---

## [Decision Letter · Decision Letter 1]

10 Aug 2021

PONE-D-20-30996R1

Effects on health-related quality of life in the randomized, controlled crossover trial ADIRA (Anti-inflammatory Diet In Rheumatoid Arthritis)

PLOS ONE

Dear Dr. Winkvist,

Thank you for submitting your manuscript to PLOS ONE. After careful consideration, we feel that it has merit but does not fully meet PLOS ONE’s publication criteria as it currently stands. Therefore, we invite you to submit a revised version of the manuscript that addresses the points raised during the review process.

We look forward to receiving your revised manuscript.

Kind regards,

Yuanyuan Wang, PhD

Academic Editor

PLOS ONE

Journal Requirements:

Additional Editor Comments (if provided):

One reviewer has made additional comments about the standard practice for analysing and reporting randomised controlled trials. I would suggest the authors revise the manuscript accordingly.

Reviewers' comments:

Reviewer's Responses to Questions

**Comments to the Author**

1. If the authors have adequately addressed your comments raised in a previous round of review and you feel that this manuscript is now acceptable for publication, you may indicate that here to bypass the “Comments to the Author” section, enter your conflict of interest statement in the “Confidential to Editor” section, and submit your "Accept" recommendation.

Reviewer #1: (No Response)

Reviewer #2: All comments have been addressed

Reviewer #3: All comments have been addressed

2. Is the manuscript technically sound, and do the data support the conclusions?

Reviewer #1: Yes

Reviewer #2: Yes

Reviewer #3: Partly

3. Has the statistical analysis been performed appropriately and rigorously? 

Reviewer #1: Yes

Reviewer #2: Yes

Reviewer #3: I Don't Know

4. Have the authors made all data underlying the findings in their manuscript fully available?

Reviewer #1: No

Reviewer #2: Yes

Reviewer #3: No

5. Is the manuscript presented in an intelligible fashion and written in standard English?

Reviewer #1: Yes

Reviewer #2: Yes

Reviewer #3: Yes

6. Review Comments to the Author

Reviewer #1: I thank the authors for their response to reviewers' comments which has vastly improved the manuscript.

There are a few remaining issues which the authors have not revised but have instead offered explanations or refutations to justify their original position. I would like to comment on these with a view to coming to an agreement that is consistent with the standard practice for analysing and reporting randomised trials.

In the results table for a continuous outcome in a randomised trial, it is standard practice to report the means and standard errors of the continuous outcome in each group - not confidence intervals - followed by the between-group differences, confidence intervals for the between-group difference, and p-values for the hypothesis tests comparing outcomes in the groups. Please amend Table 3 and Table 4 to be consistent with this standard practice.

It is also not standard practice to perform hypothesis tests/report p-values for differences in baseline characteristics between arms in a randomised trial, as you have done in Table 2. Between-group differences at baseline are not expected to occur in this study design due to randomisation, and any occurrence thereof is purely by chance and therefore there's nothing you could meaningfully make of it. It would therefore be best to remove the p-value column and note (c) from Table 2. For the same reason, it is not standard practice in randomised trials to explore confounding by baseline covariates, because randomisation 'designs away' any known/unknown/potential confounding. Arguments have been made for exploring confounding in 'small' trials to deal with chance imbalance which is more likely when you have fewer observations; however, if a study is prospectively designed to answer a research question - including a sample size calculation which justifies the number of participants required - then it cannot be deemed 'small' and susceptible to chance imbalances to justify exploring for confounding.

I had previously commented that phrases such as "a trend towards significance" or "close to significant" should be avoided, and the authors responded by saying that such small p-values, in light of the small sample size, should not lead to a conclusion of no change. The authors chose to keep these statements and to indicate that something may be going on that could be evaluated with a larger sample size/stronger study design in the future. While I agree that a conclusion of no change would not be appropriate, the correct inference to be drawn from those p-values (given the parameters of the design of the study) is "no evidence of a difference". This is not the same as concluding no change or saying that there is no difference; rather, this is simply an interpretation of the statistical evidence for/against the hypothesis being tested. After presenting the whole discussion including issues around the small size of the study, a conclusion can be drawn that bears this evidence in mind when determining what it means or does not mean. Phrases such as "a trend towards significance" or "close to significant" when interpreting hypothesis tests are poor (and non-standard) practice that should be avoided entirely.

Reviewer #2: The authors have adequately addressed all my comments and statistical analysis was appropriately performed

Reviewer #3: (No Response)

7. PLOS authors have the option to publish the peer review history of their article (what does this mean?). If published, this will include your full peer review and any attached files.

Reviewer #1: No

Reviewer #2: **Yes: **Lilia Castillo Martínez

Reviewer #3: No

---

## [Author Response · Author response to Decision Letter 1]

23 Sep 2021

Please find below our response to the comments from reviewers:

Comments from Reviewers:

Reviewer #1: I thank the authors for their response to reviewers' comments which has vastly improved the manuscript.

There are a few remaining issues which the authors have not revised but have instead offered explanations or refutations to justify their original position. I would like to comment on these with a view to coming to an agreement that is consistent with the standard practice for analysing and reporting randomised trials.

In the results table for a continuous outcome in a randomised trial, it is standard practice to report the means and standard errors of the continuous outcome in each group - not confidence intervals - followed by the between-group differences, confidence intervals for the between-group difference, and p-values for the hypothesis tests comparing outcomes in the groups. Please amend Table 3 and Table 4 to be consistent with this standard practice.

It is also not standard practice to perform hypothesis tests/report p-values for differences in baseline characteristics between arms in a randomised trial, as you have done in Table 2. Between-group differences at baseline are not expected to occur in this study design due to randomisation, and any occurrence thereof is purely by chance and therefore there's nothing you could meaningfully make of it. It would therefore be best to remove the p-value column and note (c) from Table 2. For the same reason, it is not standard practice in randomised trials to explore confounding by baseline covariates, because randomisation 'designs away' any known/unknown/potential confounding. Arguments have been made for exploring confounding in 'small' trials to deal with chance imbalance which is more likely when you have fewer observations; however, if a study is prospectively designed to answer a research question - including a sample size calculation which justifies the number of participants required - then it cannot be deemed 'small' and susceptible to chance imbalances to justify exploring for confounding.

I had previously commented that phrases such as "a trend towards significance" or "close to significant" should be avoided, and the authors responded by saying that such small p-values, in light of the small sample size, should not lead to a conclusion of no change. The authors chose to keep these statements and to indicate that something may be going on that could be evaluated with a larger sample size/stronger study design in the future. While I agree that a conclusion of no change would not be appropriate, the correct inference to be drawn from those p-values (given the parameters of the design of the study) is "no evidence of a difference". This is not the same as concluding no change or saying that there is no difference; rather, this is simply an interpretation of the statistical evidence for/against the hypothesis being tested. After presenting the whole discussion including issues around the small size of the study, a conclusion can be drawn that bears this evidence in mind when determining what it means or does not mean. Phrases such as "a trend towards significance" or "close to significant" when interpreting hypothesis tests are poor (and non-standard) practice that should be avoided entirely.

Response: Thank you so much for your engagement and your valuable inputs. We have made all the changes suggested. However, to be able to discuss and evaluate also within-group differences, we chose to keep the confidence intervals for these estimates, but as you suggested, in addition we added the standard errors of the estimates.

---

## [Editor Report · Decision Letter 2]

5 Oct 2021

Effects on health-related quality of life in the randomized, controlled crossover trial ADIRA (Anti-inflammatory Diet In Rheumatoid Arthritis)

PONE-D-20-30996R2

Dear Dr. Winkvist,

We’re pleased to inform you that your manuscript has been judged scientifically suitable for publication and will be formally accepted for publication once it meets all outstanding technical requirements.

Kind regards,

Yuanyuan Wang, PhD

Academic Editor

PLOS ONE

Additional Editor Comments (optional):

The authors have addressed the reviewer's comments properly.
---

## [Editor Report · Acceptance letter]

7 Oct 2021

PONE-D-20-30996R2 

Effects on health-related quality of life in the randomized, controlled crossover trial ADIRA (Anti-inflammatory Diet In Rheumatoid Arthritis) 

Dear Dr. Winkvist:

I'm pleased to inform you that your manuscript has been deemed suitable for publication in PLOS ONE. Congratulations! Your manuscript is now with our production department. 

Kind regards, 

on behalf of

Dr. Yuanyuan Wang 

Academic Editor

PLOS ONE